# Identification of Signatures of Positive Selection That Have Shaped the Genomic Landscape of South African Pig Populations

**DOI:** 10.3390/ani14020236

**Published:** 2024-01-12

**Authors:** Nompilo L. Hlongwane, Edgar F. Dzomba, Khanyisile Hadebe, Magriet A. van der Nest, Rian Pierneef, Farai C. Muchadeyi

**Affiliations:** 1Agricultural Research Council, Biotechnology Platform, Private Bag X5, Onderstepoort 0110, South Africa; mdladlak@arc.agric.za (K.H.); rian.pierneef@up.ac.za (R.P.); faraidzomba@gmail.com (F.C.M.); 2Discipline of Genetics, School of Life Sciences, University of KwaZulu-Natal, Private Bag X01, Scottsville 3209, South Africa; dzomba@ukzn.ac.za; 3Hans Merensky Chair in Avocado Research, Forestry and Agricultural Biotechnology Institute (FABI), University of Pretoria, Pretoria 0002, South Africa; magriet.vandernest@fabi.up.ac.za; 4Department of Biochemistry, Genetics and Microbiology, University of Pretoria, Pretoria 0002, South Africa

**Keywords:** genetic signatures, *iHS*, *XP-EHH*, *HapFLK*, pigs, gene enrichment analyses

## Abstract

**Simple Summary:**

Pigs are important in agriculture as they produce animal-based protein for human consumption. The analysis of selection signatures has implications for the maintenance and utilization of genetic diversity and can reveal genes associated with phenotypic traits, either as a result of natural or of artificial selection. Pig populations are poorly characterised in South Africa. Hence, studies aimed at evaluating genetic distinctiveness and pig breed diversity will contribute to developing a rational plan for population conservation programs among other applications.

**Abstract:**

South Africa boasts a diverse range of pig populations, encompassing intensively raised commercial breeds, as well as indigenous and village pigs reared under low-input production systems. The aim of this study was to investigate how natural and artificial selection have shaped the genomic landscape of South African pig populations sampled from different genetic backgrounds and production systems. For this purpose, the integrated haplotype score (iHS), as well as cross population extended haplotype homozygosity (XP-EHH) and Lewontin and Krakauer’s extension of the *Fst* statistic based on haplotype information (HapFLK) were utilised. Our results revealed several population-specific signatures of selection associated with the different production systems. The importance of natural selection in village populations was highlighted, as the majority of genomic regions under selection were identified in these populations. Regions under natural and artificial selection causing the distinct genetic footprints of these populations also allow for the identification of genes and pathways that may influence production and adaptation. In the context of intensively raised commercial pig breeds (Large White, Kolbroek, and Windsnyer), the identified regions included quantitative loci (QTLs) associated with economically important traits. For example, meat and carcass QTLs were prevalent in all the populations, showing the potential of village and indigenous populations’ ability to be managed and improved for such traits. Results of this study therefore increase our understanding of the intricate interplay between selection pressures, genomic adaptations, and desirable traits within South African pig populations.

## 1. Introduction

Pigs are one of the most important livestock species worldwide. In January 2020, the world population of pigs was estimated to be 677.6 million [1]. They are key for livelihoods, food security, and economic growth, especially in developing countries where they survive under harsh environments and provide for resource-limited households [2]. Besides providing proteins for humans, pigs are also used as model animals for research on human diseases [2].

Wild hog species (also referred to as wild pigs) include the warthog (*Phacochoreus Africanus*), pig deer (*Babyrousa Babyrussa*), and the pygmy hog (*Porcula Salvania*), with only wild boar (*Sus scrofa*) having been domesticated [3,4]. Changes in the phenotypic characteristics between domestic and wild pigs are highly noticeable and were driven by natural and artificial selection [5]. Independent domestication events from local wild boar in Europe and Asia gave rise to European and East Asian pigs [6,7]. As a result of strong artificial selection, there is considerable genetic distance between European and Asian domestic pigs [3,6,7]. While commercial lines of European pigs are characterised by an extended body length and lean growth, East Asian domestic pigs have good fat deposition and high reproductive performance [8,9,10].

In the absence of a reproductive barrier between East Asian and European domestic pigs, hybridisation between East Asian and European and later American pig breeds has been successfully used to increase pig production [11,12,13]. Previous studies clearly demonstrated a hybrid origin of the European Large White breed with Asian pigs [14]. The hybridisation of domesticated pigs with wild boars on European farms has also been used to increase reproduction and genetic diversity in inbred commercial pig lines [15]. Village and smallholder pigs that are farmed predominantly under free-range production systems allow for gene flow and introgression, since hybridisation occurs with wild pigs (e.g., warthogs, wild boars and bush pigs) [16,17]. Although hybridisation between domesticated and wild pigs can increase production, these events may also have a negative impact on pig production [18]. For example, it has been suggested that the outbreak of classical swine fever (CSF) is related to wild and domestic pigs mixing in free-range production systems [19,20].

There is sparse and unreliable information with regards to the history of pig populations in South Africa and other regions of Africa [21,22,23]. Indigenous breeds most likely originated from domestic pigs that spread from sub-Saharan Africa to South Africa via the Nile Corridor [22]. Commercial pig breeds from Europe and America were also introduced in South Africa for commercial farming in the 1600s by European settlers [21,22,23]. While the commercial pig breeds are known for their high performance (e.g., litter size, high growth rate, and meat and carcass quality) [24,25,26], indigenous breeds are well adapted to harsh South African environmental conditions. For example, the indigenous Windsnyer has longer black hair and a thinner epidermis for increased heat tolerance that will shield it against extreme climatic conditions [27,28]. The positive characteristics of indigenous and local populations (e.g., heat tolerance and disease resistance) are valuable and need to be characterised and conserved as they are also important to the livelihood of subsistence and small-scale farmers.

In 2013, it was estimated that South Africa had 38,500 commercial farms and 2 million smallholder farmers [29]. While commercial pig farmers practise controlled breeding and intense artificial selection for key production traits, small-scale and village farming landscapes are characterised by poorly organised and indiscriminate crossbreeding [30]. Commercial farmers mainly use European pig breeds, while pig farmers in rural areas mainly use indigenous breeds (e.g., Kolbroek and Windsnyer). However, rural farmers are increasingly shifting away from indigenous breeds towards the use of commercial exotic breeds [18,26]. Crossbreeding between exotic and indigenous breeds have been used to improve performance and production, as well as to increase tolerance and/or resistance to disease and parasites and animals that are hardy and adapted to survive under harsh local conditions [27,31,32,33].

Adaptation and domestication processes, as well as breed development, can lead to the emergence of signatures of selection in the genomes of pig populations [34]. Signatures of selection have been identified in pig populations associated with important traits, such as adaptation to high altitudes [35], muscle growth [36], and body size [10]. Genomic sequences of domestic and wild pigs have been observed to be predominantly similar, except in regions under strong selection pressure [10]. Various authors have reported on selection in domestic pigs for disease resistance, tolerance, and productivity [37,38,39]. Information on selection signatures is valuable and can be used in management strategies to improve production and adaptability. A large-scale analysis of the genetic diversity and structure of South African pig populations relative to global populations (e.g., pigs from South America, Europe, United States, and China) [40] points towards a population that has been shaped by complex evolutionary forces including domestication and continuous interactions between domestic and wild populations. This includes natural and artificial selection under the different production systems as a result of the need to adapt and survive the prevailing climatic conditions, low-input production systems, and diseases.

Many statistical methods are available to identify selection signatures. This includes the integrated haplotype score (*iHS*) that allows for the detection and characterisation of genomic regions that have experienced selection within a population [41]. The *iHS* identifies regions where a selected allele has risen in frequency quickly due to positive selection, resulting in longer haplotypes around the selected variant. Although *iHS* methods apply statistical corrections to control for confounding factors such as population structure, demographic history, or genetic drift, it is still sensitive to population structure, potentially leading to false positives. The cross-population extended haplotype homozygosity (*XP-EHH*) method takes into account differences between two populations [42]. Both tests have been shown to have high power in detecting selection signatures even in small sample sizes [43,44]. Moreover, the *XP-EHH* statistic identifies population-specific genetic signatures, by identifying regions where specific alleles or haplotypes have undergone recent positive selection, leading to their rapid increase in frequency in one population but not in another [45]. This comparison helps in differentiating between regions affected by local adaptation and those influenced by hitchhiking or a demographic history shared by multiple populations. The third approach, *hapFLK*, involves Lewontin and Krakauer’s extension of the *Fst* statistic based on haplotype information [43,46]. This test measures differences in haplotype frequencies between populations, while accounting for their hierarchical structure, enabling the capturing of population-specific genetic signatures, even in scenarios with limited sample sizes [44,47]. As a result, the *hapFLK* method is a powerful tool for identifying signatures of selection, even in the presence of bottlenecks and migration, while limiting the effects of hitchhiking.

The aim of this research was to identify and characterise genomic regions that display signatures of natural and artificial selection in South African pig populations. For this purpose, commercial, village, indigenous, wild and Vietnamese potbelly pig populations that were previously genotyped were included [40]. To improve the statistical power for detecting the selection signatures, we used the *iHS*, *XP-EHH*, and *hapFLK* approaches. Specifically, the *iHS* was used to identify and characterise signatures of selection in each of the populations, while *XP-EHH* was used to identify and characterise selection signatures between different pairs of populations. The *hapFLK* statistical method was used to identify and characterise selection signatures between the multiple populations (i.e., including all the pig populations).

## 2. Materials and Methods

### 2.1. Animal Samples, Genotyping, and Quality Control

In total, 234 animals that were previously genotyped were used in this study [40]. This included 60 pigs from commercial farms represented by the Large White (LWT), South African Landrace (SAL) and Duroc (DUR) breeds, 40 indigenous pigs represented by Kolbroek (KOL) and Windsnyer (WIN) breeds, as well as 91 village and non-descript pig populations. The latter were obtained from villages in the Eastern Cape (Alfred Nzo, ALN and O.R. Tambo, ORT) and the Limpopo (Capricorn, CAP and Mopani, MOP) districts. In addition, 5 Vietnamese Potbelly pigs (VIT) from the Johannesburg Zoo and 38 wild pigs represented by the warthog (WAT), wild boar (WBO) and bush pig (BSP) were collected from various game reserves.

The animals were genotyped using PorcineSNP60 v2 BeadChip (Illumina, San Diego, CA, USA) containing 62,163 SNPs with an average gap of 43.4 kb [40]. Markers with a call rate lower than 85% and not physically mapped to the *S. scrofa* 11.2 genome assembly were discarded using Golden Helix SNP Variation Suite (SVS) version 8.8.1. Markers with a minor allele frequency (MAF) lower than 2%, and those that deviated from the Hardy–Weinberg equilibrium (*p*-value < 0.0001) were also excluded. BEAGLE (version 5.1) was used to phase the autosomal genome using 30 iterations of the phasing algorithm on a 5 Mb chromosomal region and sample haplotype pairs for each individual per iteration for all the data sets used.

All of the pig populations (Table 1) were included in the analyses to detect selection signatures, including the populations consisting of fewer than 10 individuals (BSP, WBO, and VIT). Based on the population diversity and structure results [40], the indigenous pig populations (WIN and KOL) were grouped together (IND, *n* = 40), the village populations sampled in Limpopo (MOP and CAP) were grouped together (LIM, *n* = 52), and those in the Eastern Cape (ORT and ALN) were grouped together (EC, *n* = 39) in the *XP-EHH* and *hapFLK* analysis.

### 2.2. Detection of Signatures Using iHS

The *iHS* was used to screen for non-overlapping regions within a population under positive selection. Plink 1.9 was used to exclude duplicate SNPs and to recode all genotypes using the --allele1234 script. Plink format map and ped files were converted into the fastPHASE format using the recode fastphase script. This generated a fastphase.inp file that was used in fastPHASE software 1.4.8. This software was then used to estimate missing genotypes and unobserved haplotypes from unphased data for each chromosome. This then created an input file, the fastphase_hapguess_switch.out file, which was used to calculate the *iHS*. Once phasing was completed, the *iHS* was calculated on individual sites for possible signatures using the rehh software in the R environment. The absolute unstandardised *iHS* (un*iHS*) was identified as the log ratio *iHH^A^* ancestral to the derived *iHH^D^* allele for each SNP [41]. As the standardised *iHS* scores are roughly distributed normally with mean = 0 and standard deviation = 1, regions with an average *iHS* score of 3 (three standard deviations above the mean) or above with at least five SNPs *≤* 100 kb were considered candidate regions for selection. Manhattan plots were generated in the R package qqman 2.3.

### 2.3. Detection of Signatures Using XP-EHH

Selective sweeps between populations were detected using *XP-EHH*, which makes it possible to find selected regions using the genetic distance between adjacent SNPs based on the of *EHH* model [42]. The cross-population *EHH* (*XP-EHH*) statistic is similar to *Rsb* (Robertsonian selection bias) and compares one population to the other using haplotypes [45]. Sabeti et al. [42] defined *XP-EHH* as standardised (un*XP-EHH*), identified as a mean (un*XP-EHH*) and standard deviation (un*XP-EHH*) for each given SNP. The argument was set to pop1, identified as $p_{XP-EHH}^{right}$ relative to pop2 as $p_{XP-EHH}^{left}$. to find regions associated with each population. *XP-EHH* used the same phased file as the *iHS* did and therefore the *iHS* was firstly calculated for each population using the rehh package [47] in R. Regions with an average *XP-EHH* score of 3 (three standard deviations above the mean) or above with at least five SNPs *≤* 100 kb were considered candidate regions for selection. Manhattan plots were generated in the R package qqman.

### 2.4. Detection of Signatures Using HapFLK

To reveal genetic differentiations in genomic regions subjected to selection from multiple populations, the *HapFLK* method was employed. This test accounts for the haplotype structure of the population whilst using polymorphic SNPs in ancestral populations. Reynolds distances were calculated using *HapFLK* 1.3.0 software (https://forge-dga.jouy.inra.fr/projects/hapflk) and then converted into a kinship matrix with the HapFLK package in RStudio. A FastPHASE cross-validation procedure was used to determine haplotype diversity [46]. In total, 20 clusters with 30 maximisation iterations on a per chromosome basis were used to calculate the *HapFLK* statistic. A standard normal distribution was calculated at each SNP using *p*-values. Selected regions were identified using a *p*-value ≤ 0.001 [48]. For this study, the indigenous breed Kolbroek was used as an outgroup.

### 2.5. Annotation and Function Analyses of Identified Genomic Regions

The candidate regions identified using the three different methods (i.e., the *iHS*, *XP-EHH*, and *HapFLK*) were annotated for genes, quantitative trait loci (QTLs) and functional pathways. For this purpose, BioMart on the Ensembl gene database website was used to annotate genes at particular genome coordinates for all selected regions (release 89). A candidate region search and identification was performed within 1 Mb to the left and right of statistically significant SNPs. The current pig genome *S. scrofa* 11.2 assembly was used to extract gene symbols. Web-based Panther was employed for functional and pathway enrichment analysis. A false discovery rate (FDR) *<* 0.10 was used to assess the significance of enriched pathways. The pig QTL (Release 46) database was used to align candidate genes to an available QTL.

## 3. Results

### 3.1. Detection of Signatures within a Population Using iHS

After quality check, 27,422 SNPs in total were retained for further analysis. The *iHS* method was used to detect a positive selection within a population, and identified potential genomic regions in all 13 populations included in this study (Figure 1; Appendix A).

The number of regions identified differed greatly between the different populations ranging from 87 for CAP (Figure 1C) to 4 for BSP (Figure 1L) (Appendix A). The low numbers of regions identified in the BSP and VIT populations (Figure 1J,L) likely reflect their small sample sizes. Most selection regions were identified among the village pigs, which included the CAP (87 regions), ALN (40 regions), ORT (71 regions), and MOP (68 regions) populations (Figure 1A–D). Fewer selection regions were identified within the indigenous pig population (KOL 17; WIN 22; Figure 1H,I). In contrast to the wild boar population (WBO, 32 regions) (Figure 1M), fewer regions were identified for the warthog (WAT, 10 regions) and bush pig (BSP, 4 regions) populations (Figure 1L,M). While a high number of regions were also identified for the LWT (34 regions) and SAL (31 regions) commercial pig populations, 12 regions were identified for the DUR population (Figure 1G–I).

The regions displaying significant selection were distributed on different chromosomes, harbouring genes associated with different traits (Appendix A). For example, the region on chromosome 13 (145.36 Mbp) displaying the strongest selection signal (*iHS* score of 24.09) in the WBO population overlapped with a *GHRL* gene related to weight gain, as well as several QTLs associated with the feed conversion ratio, age at slaughter, average backfat thickness, and average daily gain, amongst others. (Figure 1M; Table 2). Other regions identified using included QTLs associated with intramuscular fat content that included the regions on chromosome 1 (ORT and CAP), chromosome 2 (LWT), chromosome 4 (KOL), chromosome 5 (WBO), chromosome 8 (WIN), chromosome 12 (ORT), and chromosome 14 (CAP), as well as QTLs associated with the number of teats that included the regions on chromosome 7 (ALN, DUR), chromosome 8 (ORT), chromosome 16 (MOP), chromosome 14 (CAP), chromosome 2 (LWT, WBO), and chromosome 14 (LWT) (Table 2).

### 3.2. Detection of Selection of Signatures between Populations Using XP-EHH

Several regions that displayed significant evidence of selection were detected between pairs of populations using *XP-EHH* (Figure 2; Appendix A). The numbers of regions displaying significant evidence of selection differed greatly between the paired populations. A high number of regions were identified between the commercial population (DUR) paired with the village (EC, 38 regions and LIM, 19 regions), indigenous (IND, 13 regions), and commercial (LWT, 23 regions) populations. Although a high number of regions were identified between the warthog population (WAT) paired with the village (LIM, 34 regions), commercial (LWT, 14 regions), and indigenous (IND, 10 regions) populations, fewer regions were identified between the WAT paired with the wild boar (WBO, 5 regions) population. This was also true for Vietnamese potbelly pigs (VIMs), with a high number of regions identified between VIM paired with the village (LIM, 14 regions and EC, 10 regions) and indigenous (IND, 11 regions) populations, while only one region was identified between VIM and the commercial (LWT, 1 regions) populations and VIMs paired with the wild boar (WBO, 5 regions) population.

Several QTLs and genes occurred in the genomic regions identified using the *XP-EHH* method (Table 3, Appendix A). The strongest signal (*XP-EHH* score of 6.91) was observed for VIT_LIM on chromosome 9 (Figure 2N). Even though this region was not associated with known QTLs, several regions identified with the *XP-EHH* method were linked with QTLs associated with important traits. For example, the regions on chromosome 1 (166.17 Mbp) and chromosome 6 (80.64 Mbp) identified for the commercial population (DUR) paired with the village populations (EC and LIM) are linked with QTLs associated with reproduction, while the region on chromosome 2 (113.82 Mbp) identified in the commercial population (DUR) paired with the village population (EC) and commercial population (LWT) is linked with QTLs associated with meat and carcass quality traits. The region identified on chromosome 1 (193.82 Mbp) detected in the wild boar population (WBO) paired with the Vietnamese potbelly pig (VIT) and commercial population (DUR) is linked to QTLs associated with key reproduction traits such as litter size, maternal infanticide, plasma droplet rate, semen volume, sperm concentration, sperm motility, and total number born alive.

### 3.3. Detection of Selection of Signatures between Populations Using HapFLK

Across all populations, regions displaying significant (*p*-value ≤ 0.001) evidence of selection were identified on chromosomes 5 and 6 using KOL as an outgroup (Figure 3).

In total, 5924 segments displaying significant (*p*-value *<* 0.10) evidence of selection were associated with 1179 genes (Appendix A). The regions on chromosomes 5 and 6 were linked with QTLs associated with intramuscular fat content, litter size, number of teats, as well as age at slaughter, meat to fat ratio, and body weight (Table 4).

### 3.4. Genes Identified Using Different Signatures of Selection Methods

The *iHS*, *XP-EHH*, and *HapFLK* methods allowed the detection of the same genomic regions on chromosomes 5 and 6. The *iHS* method detected the region on chromosome 5 in the ORT, CAP, WIN, KOL, LWT, and MOP populations, while the region on chromosome 6 was detected in DUR, ALN, ORT, and CAP. The *XP-EHH* method detected the region on chromosome 5 between the DUR_WBO pairing, while the method detected the region on chromosome 6 between the DUR_EC, DUR_WAT, and DUR_LIM pairings.

The region on chromosome 5 detected in the DUR_WBO pairing encoded *NECAP1* and *KCNJ3* genes. The GO terms reported for DUR_WBO included the regulation of ion transmembrane transport, the clathrin vesicle coat, voltage-gated potassium channel activity, the plasma membrane, vesicle-mediated transport, ligand-gated ion channel activity, and potassium transmembrane transport (Appendix A). These regions also included genes linked to important signalling pathways, namely the G-protein signalling pathway, the GABA-B_receptor_II_signalling pathway, and the muscarinic acetylcholine receptor 2 and 4 signalling pathway (Appendix A).

The genes located in the region on chromosome 6 included *EPHB2*, *EPB41L3*, *METTL4*, *EPHA8*, *LYPLA2*, *FUCA1*, *PNRC2*, *SRSF10*, *MYOM3*, *SLC16A12*, *PANK1*, and *PCGF5*. It also included important GO terms linked to the cellular response to follicle-stimulating hormone stimuli, the fucose metabolic process, the regulation of cell growth, growth factor binding, carboxylic ester hydrolase activity, and palmitoyl-(protein) hydrolase activity (Appendix A). Important pathways were found to be related to the dopamine receptor-mediated signalling pathway and Coenzyme A biosynthesis (Appendix A).

## 4. Discussion

To date, this is the first study identifying signatures of selection in South African pig populations from different genetic backgrounds. We included animals from commercial farms and villages, as well as indigenous and wild roaming pigs. The adaptation footprints across these genomic landscapes were evaluated using within and cross-population selection statistics. Although these methods accounted for small population samples, their statistical power was diminished by the small sample sizes used for bush pig, wild boar and Vietnamese potbelly pig populations [49]. Nevertheless, a number of genomic regions containing significant evidence of population-specific selection signatures were detected in the case of of wild boar, which we explored further in this study.

The population genomic approach utilised in this study allowed for the identification of genomic regions under natural selection, such as the indigenous Kolbroek and Windsyner, as well as in the wild boar. Specifically, the region on chromosome 5 included a putative *KCNJ3* gene, which may be associated with an udder structure in cattle that is typically important for production efficiencies as well as animal health and welfare [50]. This region also encoded genes involved in signalling pathways such as muscarinic acetylcholine receptors that are G protein-coupled receptors (GPCRs) playing a key role in regulating many fundamental functions (e.g., motor control, temperature control, control of inflammation, cell growth, and cell proliferation, as well as control of the airways, gastrointestinal and urinary tracts, cardiovascular system, the central nervous system, and eye) [51]. The region on chromosome 13 under natural selection in the wild boar population encoded a putative *GHRL* gene known to regulate growth and development in pigs [52,53]. The identification of these regions thus provides an opportunity to elucidate the genetic basis of the adaptive evolution of local wild and indigenous pig populations in the future including larger sample sizes.

Signatures of selection identified in the commercial pig populations included regions associated with traits such as meat and carcass quality. This is expected, as the Large White, Durocs and South African Landrace pigs are bred for meat production [5,10]. Because of this strong artificial selection and because the internal mechanism is the selection of genes, genes in these regions associated with meat and carcass quality included *CORIN* on chromosome 8, *TMPRRSS4* on chromosome 9, *SLC44A5* on chromosome 6, *APBB2* on chromosome 8, *TECTA* on chromosome 9, *LIPA* and *IDE* on chromosome 14, and *ITGA2* on chromosome 16. The *DECR1* gene on chromosome 4 is associated with cholesterol levels amongst other meat quality and growth traits. Regions associated with meat and carcass quality were also identified among the indigenous breeds. For example, the indigenous Kolbroek and Windsnyer breeds included the *JPH1* gene observed on chromosome 4, which has previously been linked to meat and carcass quality in pigs [54,55]. Furthermore, Hoffman et al. [56] observed that meat from Kolbroek pigs can be processed into bacon, ham, and chops. This shows that indigenous breeds can also be identified with traits despite their slow growth rate.

Among the genomic regions displaying signatures of selection, some were associated with fatness, an important economic trait in pig farming [57]. For example, *ITGA11* on chromosome 1 is associated with an obesity index that determines fat deposition in pigs and other animals [40]. The genomic regions (chromosomes 5 and 6) identified with all three the statistics were linked with QTLs associated with intra-muscular fat content, meat to fat ratio, and body weight. Regions that are associated with excess fat deposition when fed improved diets [36,58] present an opportunity to genetically improve meat quality in these breeds. A study by Jung et al. [59] and Ren et al. [60] observed that consumers preferred lean pork with high intramuscular fat content. As a result, commercial breeds displayed lower fat levels compared to European and Chinese breeds [61,62]. While commercial breeds (e.g., Large White, Duroc, and Landrace) have low levels of fat tissue, European breeds (e.g., Iberian and Mangalica pigs) and Chinese breeds are predisposed to accumulate excess amounts of adipose tissue [62,63,64]. Hoffman et al. [65] reported consumers’ preference towards meat with a higher lean percentage. Wild boars have low intra-muscular fat and are categorised under game meat that has high protein and iron and that is considered healthier than ordinary pork or beef meat [66,67,68]. Since pig breeds vary when it comes to fat tissue deposition with heritability levels being around 0.5 [61,62], obesity indices and intra-muscular fat can be used as potential tools for selecting animals with desirable meat and carcass qualities. For example, *SCPEP1* identified in this study regulates body fat content and correlates with intra-muscular fat deposition in pigs [69].

Genomic regions displaying signatures of selection were associated with reproduction traits such as litter size and total number born alive from a sow, semen volume, sperm concentration, sperm motility, etc. For example, *PIK3R5* is one of the genes identified in the O.R. Tambo population that influences litter size at birth and the number of piglets born alive [70], which is important as pigs differ greatly in litter size. For example, the wild boar sows an average if 6.6 litters [71] per year versus an average of 14 to 15.3 litters per sow in Large White breeds [72,73], while indigenous breeds such as Kolbroek average at 810 piglets [74]. Nowadays, the pig industry in Europe has been yielding 18–20 litters per sow [75]. This high litter number has a negative implication on the physiological tolerance for both sows and litters. The good mothering ability and hardiness of sows ensure high survival rates for the litters. Commercial breeds have an advantage being raised in the intensive production system. Several genomic regions contain QTLs associated with the number of teats on chromosomes 1, 2, 5, 6, 14, 16, and 18. The number of teats is an important trait as it ensures that piglets have adequate access to milk from the sow. The number of teats can have effects on the weaning weight of a piglet and a smaller number of teats in a sow reduces piglets’ survival rate [76]. In commercial breeds such as Large White and Duroc, a sow can have as many as 19 teats [77]. Makhanya [78] reported the number of teats to be an average of 10 in indigenous Kolbroek pigs. Various studies have shown that the number of teats is an essential morphological and reproductive trait that has been under selection for many generations in the pig industry [77,79].

A high number of regions that display significant selection were detected in the South African village pig populations. This is similar to what was previously seen for cattle, where Van Hossou et al. [80] also reported a higher number of selection signatures in admixed West African cattle populations in Benin. The presence of more selection signatures in village pig populations compared to that in other populations can be attributed to several factors. One possible explanation is that genetic diversity may provide a broader pool of genetic variants for selection to act upon, resulting in a higher number of selection signatures. Several genes related to health and resistance to parasites were identified in the village populations, which is well in line with the sturdy nature of this breed. This included the *APBB2* gene present in regions under selection in the Alfred Nzo, Mopani, and Capricorn populations, which was shown to regulate inflammatory responses during infection with porcine reproductive and respiratory syndrome virus, which is a major respiratory pathogen of pigs [71]. The *LIPA* gene under selection in the Capricorn population may be involved in the response to wounds and inflammations, as well as in the molecular genetic mechanisms affecting fecundity in sheep [72]. Village pigs are well adaptable to local harsh conditions, and this makes them important genetic resources that provide new diversity for the improvement of commercial lines.

Another explanation for the high number of regions is the admixture between village pigs and commercial pigs, which could allow for an improvement in economic traits such as reproduction, growth, and carcass traits among village pigs. Crossbreeding with commercial pigs has allowed for the introduction of genetic variants that are advantageous for these traits in not only village pigs but also indigenous pigs. For example, regions displaying selection signatures included genes for meat and carcass quality in pigs in the village (e.g., *SCPEP1* and *SAMD4A*) and indigenous (e.g., *JPH1*) populations [54,55,69]. The admixed genomes that result from the interbreeding of previously isolated populations can carry genetic signatures that resemble signals of positive selection. Therefore, the possibility that some of the genomic selection signatures identified here stem from historical admixture (i.e., they represent the “ghosts” of introgression) and not recent adaptive events could not be discounted [81]. Although further research would be needed to distinguish these types of signatures in all of the populations examined, the genetic remnants of past genetic exchange in admixed genomes may represent a valuable source of variation for further selection and/or adaptation [81].

## 5. Conclusions

This study identified several regions displaying significant signatures of selection, which are the result of natural and artificial directional selection events that have contributed to the adaptation of breeds to different environments and production systems of these pig populations. These signatures of selection allowed for the identification of the genomic regions and evolutionary processes that have shaped the populations and affect important phenotypic traits. These included traits related to reproduction, production, health, and meat and carcass quality. Meat and carcass QTLs were prevalent in all the populations, showing the potential of village and indigenous populations’ ability to be managed and improved for such traits. Our findings also confirm that genetic resources from villages and wild pigs are important for research as they are not influenced by selection when compared to commercial breeds. Additionally, as BeadChip, used in this study, may not be dense enough to fully understand the signatures between domestic and wild pigs, further research based on larger population sizes is required.

## Figures and Tables

**Figure 1 animals-14-00236-f001:**
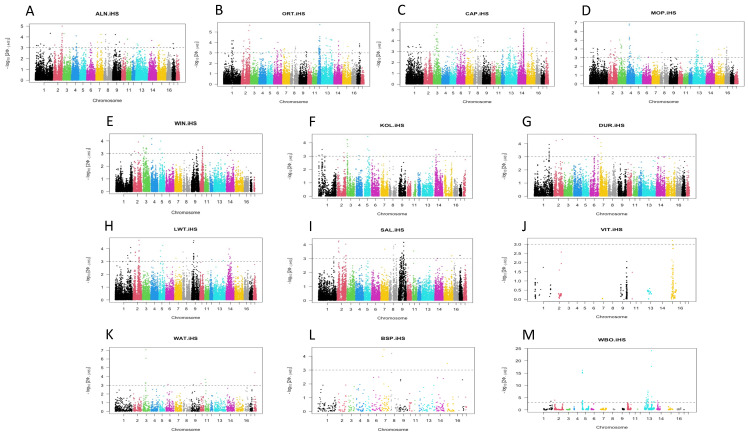
Manhattan plot of the genome-wide distribution of the selection of signatures detected via the *iHS* across the 18 chromosomes (indicated in different colours) for the village (**A**–**D**), indigenous (**E**,**F**), commercial (**G**–**I**), Vietnamese potbelly (**J**), and wild pig (**K**–**M**) populations.

**Figure 2 animals-14-00236-f002:**
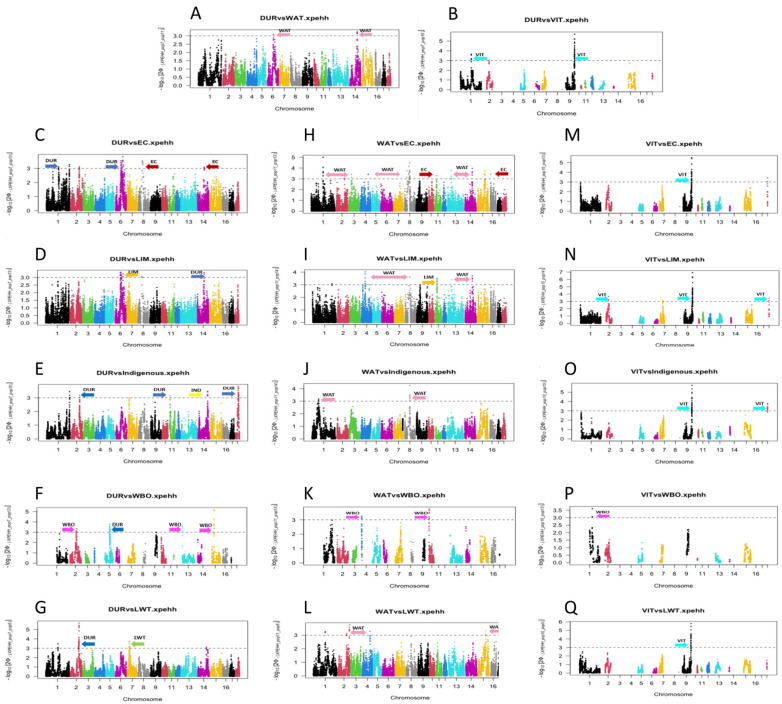
Manhattan plot of the genome-wide distribution of the selection of signatures between populations detected via *XP-EHH* across the 18 chromosomes (indicated in different colours) for DUR (**A**–**G**), WAT (**H**–**L**), and VIT (**M**–**Q**) populations.

**Figure 3 animals-14-00236-f003:**
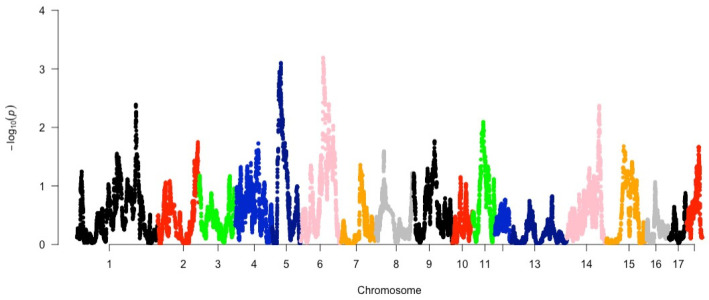
Manhattan plot for signature of selection of South African pig populations detected via *HapFLK* across 18 chromosomes (indicated in different colours).

**Table 1 animals-14-00236-t001:** Summary of the sampled pig populations.

Category	Population	Code	N
**Village**	Mopani	MOP	27
**Village**	Capricorn	CAP	25
**Village**	Oliver Reginald Tambo	ORT	22
**Village**	Alfred Nzo	ALN	17
**Commercial**	Large White	LWT	20
**Commercial**	SA Landrace	SAL	20
**Commercial**	Duroc	DUR	20
**Indigenous**	Kolbroek	KOL	20
**Indigenous**	Windsnyer Type	WIN	20
**Asian**	Vietnamese Potbelly	VIT	5
**Wild**	Wild boar	WBO	4
**Wild**	Wild boar	WAT	31
**Wild**	Wild boar	BSP	3

**Table 2 animals-14-00236-t002:** Within-population list of genomic regions under selection and candidate genes detected using the *iHS* method.

Populations	Chr	Start	End	Gene	QTLs
**ALN**	2	25247505	25378565	*TRIM44*	Spinal Curvature
	7	95982739	96254303	*DPF3*	Teat number
	8	31921587	32301975	*APBB2*	Stearic acid content
	13	29753308	29775371	*PTH1R*	Front leg conformation, Hind leg conformation, Hip structure
**ORT**	1	162364055	162737492	*NEDD4L*	Intramuscular fat content
	2	87678811	87866856	*DMGDH*	Litter weight piglets born alive
	8	80290715	80669404	*NR3C2*	Loin muscle, Teat number
	12	53979215	54055377	*PIK3R5*	Intramuscular fat content
	12	27957732	28617585	*CA10*	Hemaglobin
	12	44617499	44660271	*VTN*	Body depth, Drip loss, Hind leg conformation, Hip structure, pH 24 h post-mortem (loin), ph 45 min post-mortem
**MOP**	4	71045333	71318030	*NKAIN3*	Body weight (birth)
	8	31921587	32301975	*APBB2*	Stearic acid content
	16	27537885	27631130	*SELENOP*	Meat colour a
	16	32142034	32148335	*PELO*	Obesity index, Teat number
	16	32336292	32437103	*ITGA2*	Body weight (5 weeks)
**CAP**	1	183915339	184140122	*SAMD4A*	Intramuscular fat content
	12	27416337	27436160	*NME1*	Conductivity 24 h post-mortem (loin), Cooking loss, Loin muscle depth, Loin weight
	12	27957732	28617585	*CA10*	Hemoglobin
	14	101123381	101254725	*LIPA*	Average daily gain, Front leg weight, HDL/LDL ratio, Litter size, Loin muscle area, Monounsaturated fatty acid content, Oleic acid content, Skin thickness, Sperm concentration, Teat number
	14	103992037	104106576	*IDE*	e.g., Abdominal fat percentage, Age at puberty, Age at slaughter, Average backfat thickness, Average daily gain, Backfat at first rib, Backfat at last rib, Backfat between 3rd and 4th last ribs, Body depth, Body height
	14	105036770	105044765	*RBP4*	Litter size, Total number born alive
	14	12437515	12563544	*EXTL3*	Fat androstenone level
	8	76482022	76699351	*FBXW7*	Body mass index
	8	31921587	32301975	*APBB2*	Stearic acid content
**LWT**	2	108255930	108536881	*PAM*	Intramuscular fat content, Loin percentage, Maternal infanticide, Teat number
	7	26860140	26990017	*LRRC1*	Meat colour L
	9	48120802	48286278	*TECTA*	Backfat between 3rd and 4th last ribs
	13	177365717	179013542	*ROBO2*	Feed efficiency, Linolenic acid content
	14	69200215	70938204	*CTNNA3*	Teat number
**DUR**	6	137595524	138010444	*SLC44A5*	Diameter of muscle fibers
	7	27389874	27895570	*KHDRBS2*	Loin muscle area, Loin muscle depth, Teat number
**KOL**	4	61628299	61716546	*JPH1*	Intramuscular fat content
	14	79352396	80106258	*KCNMA1*	Meat colour b
**WIN**	5	18718158	18723843	*TARBP2*	Backfat between 3rd and 4th last ribs
	8	127731903	128953331	*CCSER1*	Backfat between 3rd and 4th last ribs, Intramuscular fat content
**VIT**	2	79766349	80141293	*COL23A1*	Front foot size, Hip structure
	4	79687359	79847281	*PRKDC*	Feed conversion ratio
	6	79849687	79958271	*HSPG2*	Days to 113 kg, Marbling
	14	79352396	80106258	*KCNMA1*	Meat colour b
**WAT**	18	31027031	31125465	*MDFIC*	Fat androstenone level
**WBO**	2	15791451	15819138	*F2*	e.g., Age at slaughter, Average backfat thickness, Average daily gain, Backfat at last rib, Backfat at rump, Backfat thickness between 3rd and 4th rib, Body weight (end of test), Body weight (weaning)
	5	4769801	4849334	*SHISAL1*	Intramuscular fat content
	5	6997444	7014422	*POLR3H*	Fat androstenone level
	13	66316436	66452917	*GHRL*	Age at slaughter, Average daily gain, Daily feed intake, Days to 100 kg, Feed intake, Loin weight, Marbling
	14	21754463	22121051	*SPOCK3*	Fat androstenone level

**Table 3 animals-14-00236-t003:** Selected regions and candidate genes detected between pairs of populations using the *XP-EHH* method.

Populations	Chr	Start	End	Gene	QTLs
**DUR_EC**	1	166173135	166310972	*ITGA11*	Obesity index, Teat number
	2	113774412	114206930	*FER*	Abdominal circumference, Average backfat thickness, Average daily gain, Backfat at last lumbar, Biceps brachii weight, Body height, Body weight (3 weeks), Carcass weight (hot), Double-bond index
	6	80649143	80843567	*EPHB2*	Litter weight total
**DUR_LIM**	1	166173135	166310972	*ITGA11*	Obesity index, Teat number
	6	80649143	80843567	*EPHB2*	Litter weight total
**DUR_IND**	18	40820644	41409087	*PDE1C*	Backfat at rump
	18	42030510	42046184	*GHRHR*	Backfat at last rump, Carcass length, Fat-cuts percentage
**DUR_LWT**	2	113774412	114206930	*FER*	e.g., Abdominal circumference, Arachidonic acid content, Aspartate aminotransferase activity, Average backfat thickness, Average daily gain, Backfat at last lumbar, Backfat at mid-back, Backfat at rump, Backfat at tenth rib
	7	27389874	27895570	*KHDRBS2*	Loin muscle area, Loin muscle depth, Teat number
	7	30708332	30724045	*SNRPC*	Loin muscle area, Loin muscle depth
	7	30731461	30802735	*UHRF1BP1*	Femur length, Hip bone length, Humerus length, Tibia length, Ulna length
	7	30812920	30995361	*ANKS1A*	Femur length, Hip bone length, Humerus length, Tibia length, Ulna length, Galt score (front), Loin muscle area, Loin muscle depth
	7	31586555	31603617	*ARMC12*	Facial morphology
	7	31722990	31792904	*SLC26A8*	Facial morphology, Femur length, Humerus length, Tibia length, Ulna length
	14	111834168	111914304	*PAX2*	Monounsaturated fatty acid to saturated fatty acid ratio, Oleic acid to stearic acid ratio, Palmitoleic acid to palmitic acid ratio, Stearic acid content
**DUR_VIT**	2	72326714	72391648	*VAV1*	Average daily gain, Backfat between 3rd and 4th last rib, Birth weight variability, Body weight (end of test), Conductivity 45 min post-mortem, Fat androstenone level, Intramuscular fat content, Time in feeder per day, pH 24 h postmortem (ham), pH 45 min postmortem
**DUR_WBO**	1	193722164	193906565	*ESR2*	Front leg conformation, Gait score (overall), Hind leg conformation, Litter size, Maternal infanticide, Plasma droplet rate, Semen volume, Sperm concentration, Sperm motility, Total number born alive
**WAT_EC**	1	254683885	254703225	*AMBP*	Conductivity 24 h post-mortem (loin), pH 24 h postmortem (ham), pH 24 h post-mortem (loin), pH 45 min postmortem
	3	39957269	39957727	*NPW*	Lean meat percentage
	8	37530815	37809761	*CORIN*	Platelet count
	8	37797875	37875540	*NFXL1*	Mean corpuscular hemoglobin content, Mean corpuscular volume
	8	47473787	47601359	*RXFP1*	Red blood cell count
	8	71520275	71554766	*PPEF2*	Platelet distribution width
	8	71573783	71603338	*NAAA*	Platelet distribution width
	8	72543620	72679173	*SEPTIN11*	Teat number
	8	73502664	73958083	*FRAS1*	Teat number
**WAT_LIM**	1	254683885	254703225	*AMBP*	Conductivity 24 h post-mortem (loin), pH 24 h postmortem (ham), pH 24 h post-mortem (loin), pH 45 min postmortem
	8	71520275	71554766	*PPEF2*	Platelet distribution width
	8	71573783	71603338	*NAAA*	Platelet distribution width
	14	113414264	113429936	*PSD*	Intramuscular fat content, Oleic acid to stearic acid ratio
	14	113464174	113478699	*MFSD13A*	Oleic acid content, Oleic acid to stearic acid ratio, Stearic acid content
	14	113480197	113498740	*ACTR1A*	Oleic acid content, Stearic acid content
	9	45400464	45435916	*TMPRSS4*	Backfat between 3rd and 4th last ribs
**WAT_LWT**	4	105804586	105845725	*CSDE1*	Intramuscular fat content
	4	105868897	105893771	*AMPD1*	Juiciness score, Overall impression, sensory panel, Tenderness score
**WAT_WBO**	7	89120822	89168270	*MAX*	Meat colour b, Teat number, maximum per side
**VIT_EC**	18	29895878	29936233	*TES*	Average daily gain, Backfat between 3rd and 4th last rib, Birth weight variability, Body weight (end of test), Conductivity 45 min post-mortem, Fat androstenone level, Intramuscular fat content, Time in feeder per day, pH 24 h postmortem (ham), pH 45 min postmortem, Teat number
**VIT_LIM**	2	72326714	72391648	*VAV1*	Average daily gain, Backfat between 3rd and 4th last rib, Birth weight variability, Body weight (end of test), Conductivity 45 min post-mortem, Fat androstenone level, Intramuscular fat content, Time in feeder per day, pH 24 h postmortem (ham), pH 45 min postmortem
	18	29895878	29936233	*TES*	Average daily gain, Backfat between 3rd and 4th last rib, Birth weight variability, Body weight (end of test), Conductivity 45 min post-mortem, Fat androstenone level, Intramuscular fat content, Time in feeder per day, pH 24 h postmortem (ham), pH 45 min postmortem, Teat number
**VIT_IND**	18	29895878	29936233	*TES*	Average daily gain, Backfat between 3rd and 4th last rib, Birth weight variability, Body weight (end of test), Conductivity 45 min post-mortem, Fat androstenone level, Intramuscular fat content, Time in feeder per day, pH 24 h postmortem (ham), pH 45 min postmortem, Teat number
	18	31027031	31125465	*MDFIC*	Fat androstenone level
**VIT_WBO**	1	193722164	193906565	*ESR2*	Front leg conformation, Gait score (overall), Hind leg conformation, Litter size, Maternal infanticide, Plasma droplet rate, Semen volume, Sperm concentration, Sperm motility, Total number born alive

**Table 4 animals-14-00236-t004:** Genomic regions under selection detected via *HapFLK* methods in South African pigs.

Chr	Start	End	Gene	QTLs
5	44381009	44558744	*FAR2*	Feed conversion ratio
5	56817606	57005114	*EPS8*	Fat androstenone level
5	64519186	65002098	*VWF*	Litter size
5	66665263	66838922	*PRMT8*	Teat number
5	28300950	28605120	*SRGAP1*	Ear area
5	29695839	29863599	*MSRB3*	Ear area
5	97092883	97148601	*SLC6A15*	Time in feeder per day
5	33858572	34033939	*CCT2*	Feed conversion ratio
5	34067992	34218513	*MYRFL*	Feed conversion ratio
5	34660029	34794321	*PTPRB*	Feed conversion ratio
5	36274364	36658703	*TRHDE*	Feed conversion ratio
6	97342474	97429364	*GNAL*	Age at puberty, Arachidic acid content, Average backfat thickness, Average daily gain, Backfat at last lumbar, Backfat at last rib, Backfat at rump, Backfat at tenth rib, Body weight (16 days), Carcass weight (hot), ear area, Feed conversion ratio, Lean meat percentage, Loin muscle area, Loin muscle depth, Oleic acid content, Oleic acid to stearic acid ratio, PH for longissmus dorsi, Stearic acid content, Teat number, Vertebra number, Androstenone laboratory
6	108115886	108227748	*CABLES1*	Average daily gain, Backfat at rump
6	108548837	108805693	*LAMA3*	Average daily gain
6	75696484	75755892	*PADI2*	Fat androstenone level
6	112396721	112628432	*CDH2*	Average daily gain, Intramuscular fat content, Lean meat percentage, Obesity index
6	117631227	118334105	*NOL4*	Average backfat thickness
6	79849687	79958271	*HSPG2*	Days to 113 kg, Marbling
6	125890708	126043161	*PIK3C3*	Average backfat thickness, Average daily gain, Intramuscular fat content, Loin muscle area
6	80649143	80843567	*EPHB2*	Litter weight (total)
6	137595524	138010444	*SLC44A5*	Diameter of muscle fibers
6	43933759	44069330	*GPI*	Average backfat thickness, Body weight (5 weeks), Intramuscular fat content, Osteochondrosis score
6	46442857	46470075	*ZNF570*	Lean meat percentage
14	47946396	48040822	*LIMK2*	Fat androstenone level, Melanoma susceptibility
14	122777130	122828051	*ACSL5*	Fat androstenone level
14	123343694	123546417	*TCF7L2*	Carcass weight (hot), Number of visits to feeder per day
14	133460167	133544018	*CHST15*	Teat number
14	124760398	125010892	*ABLIM1*	Fat androstenone level, Intramuscular fat content
15	100868469	101211242	*ANKRD44*	Skin thickness
15	101623818	101957516	*PLCL1*	Skin thickness
15	118335925	118452296	*XRCC5*	Average backfat thickness, Conductivity 24 h post-mortem (loin), Cooking loss, Fat weight (total), Lean meat percentage, Loin muscle area, Loin muscle depth, Loin weight, PH for longissmusdorsi, Subcutanous fat area, pH 24 h postmortem (ham), pH 24 h post-mortem (ham), pH 24 h post-mortem (loin)
15	79935994	80025850	*SP3*	Cooking loss, Meat colour b, Shear force, Thawing loss
16	32130235	32304592	*ITGA1*	Obesity index, Teat number
16	32336292	32437103	*ITGA2*	Body weight (5 weeks)
16	33126494	33565542	*ARL15*	Loin muscle area
18	34006688	34906268	*IMMP2L*	Age at puberty
18	40820644	41409087	*PDE1C*	Backfat at rump
18	42030510	42046184	*GHRHR*	Backfat at last rib, Carcass length, Fat-cuts percentage
18	51387836	51802945	*HECW1*	Teat number

## Data Availability

The datasets that were analysed during the current study are available from the corresponding author upon reasonable request.

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
