# Peer review of "Identification of Signatures of Positive Selection That Have Shaped the Genomic Landscape of South African Pig Populations"

_animals, 2024, doi:10.3390/ani14020236_

Round 1
Reviewer 1 Report
Comments and Suggestions for Authors
The title reflects the major findings of the work and well represents the study.
Please, avoid the use of personal form (we, our...) throughout the text.
The abstract adequately summarize methodology and results. However, Authors should correct English language (an example is the sentence, Lines 24-26 “The aim of this study was therefore to investigated how natural and artificial selection have shaped the genomic landscape of South African pig populations sampled from diverse genetic backgrounds and production systems”, please correct as “The aim of this study was, therefore, to investigate how natural and artificial selection have shaped the genomic landscape of South African pig populations sampled from different genetic backgrounds and production systems”).
Please, improve English language throughout the text.
The statistical analysis applied on data should be indicated in this section as well as the P values for the results.
The introduction section is well written and it falls within the topic of the study.
In the sentence (Lines 105-106) “Various authors have reported on selection in domestic pigs for resistance and tolerance (40) and productivity (41).” I suggest to add the recent papers “D’Alessando E. et al., Frontiers in Veterinary Science 9, 2022,1046101; Sutera A.M. et al., Animals 2023, 13, 1750; Floridia V: et al., Animals 2023, 13, 2323.”
The section of Materials and Methods is clear for the reader and adequately describes the methods applied in the study.
Results section as well as Discussion section is clear and well written. The findings obtained in the study were well discussed and justified with appropriate references.
The conclusion section is clear and well written. Authors well summarize the results of the study and well emphasize the significance of the survey.
The figurer are generally good. Please, consider to improve the presentation of Tables.
Comments on the Quality of English LanguageThe English language should be improved throughout the text.
Author Response
Reviewer 1
Please, avoid the use of personal form (we, our...) throughout the text.
Response: The recommended revisions were made. See Lines 28, 29, 376-379 and 476.
The statistical analysis applied on data should be indicated in this section, as well as the P values for the results.
Response: As suggested, the required information is included in the methods section.
Authors should correct English language (an example is the sentence, Lines 24-26 “The aim of this study was therefore to investigated how natural and artificial selection have shaped the genomic landscape of South African pig populations sampled from diverse genetic backgrounds and production systems”, please correct as “The aim of this study was, therefore, to investigate how natural and artificial selection have shaped the genomic landscape of South African pig populations sampled from different genetic backgrounds and production systems”).
Response: The recommended revisions were made. See Lines 24-26.
In the sentence (Lines 105-106) “Various authors have reported on selection in domestic pigs for resistance and tolerance (40) and productivity (41).” I suggest to add the recent papers “D’Alessando E. et al., Frontiers in Veterinary Science 9, 2022,1046101; Sutera A.M. et al., Animals 2023, 13, 1750; Floridia V: et al., Animals 2023, 13, 2323.”
Response: The recommended revisions were made.
Please, consider to improve the presentation of Tables.
Response: The recommended revisions were made.
Please, improve English language throughout the text.
Response: The recommended revisions were made.
Reviewer 2 Report
Comments and Suggestions for Authors
This study from Hlongwanel named “Identification of signatures of selection of and affected QTLs in pigs through genomic scans using different methods” investigated the selection signals of South African pigs, and identified several genes and QTLs related to important economic traits. However, there are numerous issues with the current draft, and the following suggestions are offered for consideration:
1.The manuscript discussed multiple breeds, including commercial and indigenous pigs, and employed various methods to detect selection signatures. However, the results appear scattered overall and do not form cohesive whole research, nor do they present a clear and definitive conclusion.
2. The analysis of population structure is missing. In general, principal component analysis, phylogenetic trees, and population structure need to be analyzed. If possible, South African indigenous pig breeds could be marked on the map.
3. After quality control, only 27,740 SNPs were available for analysis, which is too low a marker density. This has resulted markers distributed only on few chromosomes in some populations, rendering the results unreliable. It should be clarified how many SNPs were excluded by each quality control condition.
3. The figures in the manuscript are disorganized and need to be rearranged.
4. The abstract needs to be rewritten as most of the results are not reflected in it.
5. The title is not smooth, and individual sentences are incomplete, such as line 97.
6. Which version of the reference genome was used? Sus scrofa 11.1 (line 205) or Sus scrofa 11.2 (lines 147-148)?
Author Response
Reviewer 2
The title is not smooth. “Identification of signatures of selection and affected QTLs in pigs through genomic scans using different methods”
Response: We agree with the reviewer. The title of the revised manuscript now reads “Identification of signatures of positive selection that have shaped the genomic landscape of South African pig populations.”
The abstract needs to be rewritten as most of the results are not reflected in it.
Response: The recommended revisions were made.
The manuscript discussed multiple breeds, including commercial and indigenous pigs, and employed various methods to detect selection signatures. However, the results appear scattered overall and do not form cohesive whole research, nor do they present a clear and definitive conclusion.
Response: The recommended revisions were made.
After quality control, only 27,740 SNPs were available for analysis, which is too low a marker density. This has resulted markers distributed only on few chromosomes in some populations, rendering the results unreliable. It should be clarified how many SNPs were excluded by each quality control condition.
Response: This study included 234 animals previously genotyped using the PorcineSNP60 v2 genotyping BeadChip (Illumina, United States) containing 62,163 SNPs that were developed using five pig breeds (Duroc, Pietrain, Landrace, Large White and European Wild Boar (Ramos et al., 2008). Of the 62,163 SNPs, 45,942 were polymorphic when tested among European Wild Boar animals. Warthog, Bush pig and wild pigs that are native to Africa are often geographically restricted and harbour unique genetic variants (Hlongwane et., 2020). However, this variation was not included when developing the PorcineSNP60 v2 genotyping BeadChip. This is the reason why only 27,740 SNPs were available for analysis, although the threshold of deviation from Hardy-Weinberg Equilibrium was set to P<0.0001.
The analysis of population structure is missing. In general, principal component analysis, phylogenetic trees, and population structure need to be analyzed.
Response: These analyses have already been published (Hlongwane et al., 2020).
If possible, South African indigenous pig breeds could be marked on the map.
Response: The recommended revisions were made.
The figures in the manuscript are disorganized and need to be rearranged.
Response: The Figures were combined and/or rearranged in the revised manuscript.
Which version of the reference genome was used? Sus scrofa 11.1 (line 205) or Sus scrofa 11.2 (lines 147-148)?
Response: In the revised manuscript Sus scrofa 11.1 was changed to Sus scrofa 11.2 See Line 205.
Individual sentences are incomplete, such as line 97.
Response: This incomplete sentence was removed from the revised manuscript.
Reviewer 3 Report
Comments and Suggestions for Authors
Lines 97 and 98: the sentence is interrupted by a full stop. Its seems like the authors wanted it to be one sentence but its not clear. Please rectify.
Line 111: add a coma after “To date”
Line 113: points instead of point since analysis is singular
Line 114: use an “and” instead of the comma.
Line 236: change “are” to “were”
Line 312: change “found” to “considered”
Line 355: reported diverse what? (incomplete thought here). Please complete.
Line 362: Do you meant within DUR_WBO or between it and some other breed?
Line 368: add a comma before “allowed”
Line 347: affect not affects as processes are plural.
Line 384: replace under with “of”. Alternatively, you can say regions under selection.
Line 388: remove “that”
Lines 401 and 402: the sentence starting “Research done by….” The sentence seems to incomplete and I couldn’t understand what the authors were getting at. Please revisit and clarify.
Line 426: chromosome xx?
Comments on the Quality of English Language
Lines 97 and 98: the sentence is interrupted by a full stop. Its seems like the authors wanted it to be one sentence but its not clear. Please rectify.
Line 111: add a coma after “To date”
Line 113: points instead of point since analysis is singular
Line 114: use an “and” instead of the comma.
Line 236: change “are” to “were”
Line 312: change “found” to “considered”
Line 355: reported diverse what? (incomplete thought here). Please complete.
Line 362: Do you meant within DUR_WBO or between it and some other breed?
Line 368: add a comma before “allowed”
Line 347: affect not affects as processes are plural.
Line 384: replace under with “of”. Alternatively, you can say regions under selection.
Line 388: remove “that”
Lines 401 and 402: the sentence starting “Research done by….” The sentence seems to incomplete and I couldn’t understand what the authors were getting at. Please revisit and clarify.
Line 426: chromosome xx?
Author Response
Reviewer 3
Lines 97 and 98: the sentence is interrupted by a full stop. Its seems like the authors wanted it to be one sentence but its not clear. Please rectify.
Line 111: add a coma after “To date”
Line 113: points instead of point since analysis is singular
Line 114: use an “and” instead of the comma.
Line 236: change “are” to “were”
Line 312: change “found” to “considered”
Line 355: reported diverse what? (incomplete thought here). Please complete.
Line 362: Do you meant within DUR_WBO or between it and some other breed?
Line 368: add a comma before “allowed”
Line 374: affect not affects as processes are plural.
Line 384: replace under with “of”. Alternatively, you can say regions under selection.
Line 388: remove “that”
Lines 401 and 402: the sentence starting “Research done by….” The sentence seems to incomplete and I couldn’t understand what the authors were getting at. Please revisit and clarify.
Line 426: chromosome xx?
Response: The recommended revisions were made.
Reviewer 4 Report
Comments and Suggestions for Authors
This manuscript investigates the selection signatures of different pig populations, including commercial, village, indigenous, and wild pigs in South Africa, using three population genetics approaches. All pigs used in this manuscript were previously genotyped using the PorcineSNP60 v2 genotyping BeadChip. The sample size in each pig population is around 20 pigs, which would be very small for this kind of analysis. Even if QTLs for important traits that have been previously identified in different studies are located on the signatures of this study, it does not necessarily mean that these QTLs affect the corresponding traits of the pig populations examined in this study. Furthermore, the manuscript is challenging to understand. Hence, most of the results obtained from the signatures may be estimations, not based on the actual data obtained from the pigs used in this study.
My comments are shown below.
1. Introduction: The authors used the same pig populations and the same SNPs. It should clarify the difference between this study and your previous study (Ref. 43) in the Introduction.
2. Lines 97-98: This sentence is not completed.
3. Materials and Methods: Lines 137-144: No actual number of pigs in each population is provided in the text at all. It should be shown in this manuscript. How were heterozygotes for SNPs treated in this study?
4. According to the authors’ previous study (Ref. 43), the sample size in each population ranged from 15 to 28 pigs, and these pigs were again used in this study. In contrast, the Ref. 43 study estimated the effective population size of each population, which ranged from 34 to 271 pigs at Generation 12 (Table S1 of Ref. 43). These effective population sizes are much greater than the pig sizes used in this study. Using such a small sample size in each population, why can signatures within a population be estimated? The small sample size in each population would not reflect all the genetic variation in the population. Hence, the three analyses, iHS and XP-EHH and HapFLK, might not work adequately.
5. Line 150: The threshold of deviation from Hardy-Weinberg Equilibrium is set to P<0.0001. This threshold is lower than thresholds for iHS and XP-EHH (both P<0.001) and HapFLK (P<0.10). Is there the possibility that SNPs at the thresholds between P<0.0001 and P<0.001, which do not deviate from Hardy-Weinberg Equilibrium, are identified as ghost SNPs in iHS, XP-EHH, and HapFLK analyses? In other words, to prevent ghost SNPs as little as possible, why is a much higher threshold (e.g., P<0.05) used for the deviation testing from Hardy-Weinberg Equilibrium?
6. Lines 206-209: No methods or criteria for the detection of candidate genes and QTLs are explained.
7. Lines 230-238: It is hard to understand the contents of this paragraph. Please rephrase with an easy-to-understand explanation for each population.
8. Lines 240-249: No trait data, such as litter size and stearic acid content in the pig populations used, are shown in the manuscript. However, for example, the authors mentioned that genes such as LIPA (101.13 Mbp), IDE (103.99 Mbp), and RBF4 (105.04 Mbp) on chromosome 4 in the CAP population were associated with litter size. The authors should show whether such associations are present or not using the authors’ own trait data.
Author Response
Reviewer 4
The sample size in each pig population is around 20 pigs, which would be very small for this kind of analysis. Even if QTLs for important traits that have been previously identified in different studies are located on the signatures of this study, it does not necessarily mean that these QTLs affect the corresponding traits of the pig populations examined in this study.
Response: In the revised manuscript it is more clearly explained why the test used can be empployed for the identification of regions displaying signatures of selection. See lines 126-135 that now reads: “Both tests have been shown to have high power in detecting selection signatures even in small sample sizes (Sabeti et al., 2002; Eydivandi et al., 2021). Moreover, the XP-EHH statistic assesses haplotype differences between two populations and is calculated for each population pair, indicating its applicability in capturing population-specific genetic signatures (Chen et al., 2023)”.
Furthermore, the manuscript is challenging to understand. Hence, most of the results obtained from the signatures may be estimations, not based on the actual data obtained from the pigs used in this study.
Response: The results section was revised to addressed this issue.
The authors used the same pig populations and the same SNPs. It should clarify the difference between this study and your previous study (Ref. 43) in the Introduction.
Response: The recommended revisions were made. See Lines 109-111.
Lines 97-98: This sentence is not completed.
Response: The incomplete sentence was removed from the revised manuscript.
Lines 137-144: No actual number of pigs in each population is provided in the text at all. It should be shown in this manuscript.
Response: The recommended revisions were made. See Lines 133-142.
According to the authors’ previous study (Ref. 43), the sample size in each population ranged from 15 to 28 pigs, and these pigs were again used in this study. In contrast, the Ref. 43 study estimated the effective population size of each population, which ranged from 34 to 271 pigs at Generation 12 (Table S1 of Ref. 43). These effective population sizes are much greater than the pig sizes used in this study. Using such a small sample size in each population, why can signatures within a population be estimated? The small sample size in each population would not reflect all the genetic variation in the population. Hence, the three analyses, iHS and XP-EHH and HapFLK, might not work adequately.
Response: We agree with the reviewer, the small sample size in each population may not fully capture the genetic variation present, potentially leading to a lack of representation of the entire population's genetic diversity. While the small sample size in each population may not fully reflect all genetic variation, the HIS, XP-EHH and HAPFLK tests been have demonstrated to have high power in detecting selection signatures, making them suitable methods for identifying recent positive selection even in scenarios with limited sample sizes (Lines 123-130). Furthermore, selective sweeps, which can increase genetic differentiation among populations, have been shown to cause allele frequency spectra to depart from the expectation under neutrality. This suggests that even with small sample sizes, the effects of positive selection on genetic variation can still be detected. Moreover, the XP-EHH statistic assesses haplotype differences between two populations and is calculated for each population pair, indicating its applicability in capturing population-specific genetic signatures.
Line 150: The threshold of deviation from Hardy-Weinberg Equilibrium is set to P<0.0001. This threshold is lower than thresholds for iHS and XP-EHH (both P<0.001) and HapFLK (P<0.10). Is there the possibility that SNPs at the thresholds between P<0.0001 and P<0.001, which do not deviate from Hardy-Weinberg Equilibrium, are identified as ghost SNPs in iHS, XP-EHH, and HapFLK analyses? In other words, to prevent ghost SNPs as little as possible, why is a much higher threshold (e.g., P<0.05) used for the deviation testing from Hardy-Weinberg Equilibrium?
Response: This study included 234 animals previously genotyped using PorcineSNP60 v2 genotyping BeadChip (Illumina, United States) containing 62,163 SNPs that were develop using five pig breeds (Duroc, Pietrain, Landrace, Large White and European Wild Boar (Ramos et al., 2008). Of the 62,163 SNPs, 45,942 were polymorphic when tested among European Wild Boar animals*. Warthog, Bush pig and wild pigs that are native to Africa are often geographically restricted, harboring unique genetic variants (Hlongwane et., 2020). However, this variation was not included when developing the PorcineSNP60 v2 genotyping BeadChip. Because of this, only 27,740 SNPs were available for analysis, although the threshold of deviation from Hardy-Weinberg Equilibrium was set to P<0.0001.
*(https://www.illumina.com/content/dam/illumina-marketing/documents/products/datasheets/datasheet_porcinesnp60.pdf)
Round 2
Reviewer 2 Report
Comments and Suggestions for Authors
I have no more comments for this version.
In the Reference, the name of the journals is the full name or abbreviation, it needs to be written in accordance with the requested of the journal.
Author Response
Reviewer 2:
In the Reference, the name of the journals is the full name or abbreviation, it needs to be written in accordance with the requested of the journal.
Response: All references were checked and revised.
Reviewer 4 Report
Comments and Suggestions for Authors
Furthermore, the manuscript is challenging to understand. Hence, most of the results obtained from the signatures may be estimations, not based on the actual data obtained from the pigs used in this study.
Response: The results section was revised to addressed this issue.
Question: Unclear. In other words, how did the authors exclude hitchhiking effects in the signatures? That is, some of the likely genes shown in the tables may include genes that are not related to population differences.
Lines 137-144: No actual number of pigs in each population is provided in the text at all. It should be shown in this manuscript.
Response: The recommended revisions were made. See Lines 133-142.
Question: No numbers are shown in the lines. Again, please indicate the number of pigs used in each population such as LWT, SAL, ALN, ORT, MOP, etc.
Response: We agree with the reviewer, the small sample size in each population may not fully capture the genetic variation present, potentially leading to a lack of representation of the entire population's genetic diversity. While the small sample size in each population may not fully reflect all genetic variation, the HIS, XP-EHH and HAPFLK tests been have demonstrated to have high power in detecting selection signatures, making them suitable methods for identifying recent positive selection even in scenarios with limited sample sizes (Lines 123-130). Furthermore, selective sweeps, which can increase genetic differentiation among populations, have been shown to cause allele frequency spectra to depart from the expectation under neutrality. This suggests that even with small sample sizes, the effects of positive selection on genetic variation can still be detected. Moreover, the XP-EHH statistic assesses haplotype differences between two populations and is calculated for each population pair, indicating its applicability in capturing population-specific genetic signatures.
Question: This is true only for the pigs used in the study. However, it would not be possible to detect all effects of positive selection. An explanation would need to be made for the small sample size in the Discussion.
Line 150: The threshold of deviation from Hardy-Weinberg Equilibrium is set to P<0.0001. This threshold is lower than thresholds for iHS and XP-EHH (both P<0.001) and HapFLK (P<0.10). Is there the possibility that SNPs at the thresholds between P<0.0001 and P<0.001, which do not deviate from Hardy-Weinberg Equilibrium, are identified as ghost SNPs in iHS, XP-EHH, and HapFLK analyses? In other words, to prevent ghost SNPs as little as possible, why is a much higher threshold (e.g., P<0.05) used for the deviation testing from Hardy-Weinberg Equilibrium?
Response: This study included 234 animals previously genotyped using PorcineSNP60 v2 genotyping BeadChip (Illumina, United States) containing 62,163 SNPs that were develop using five pig breeds (Duroc, Pietrain, Landrace, Large White and European Wild Boar (Ramos et al., 2008). Of the 62,163 SNPs, 45,942 were polymorphic when tested among European Wild Boar animals*. Warthog, Bush pig and wild pigs that are native to Africa are often geographically restricted, harboring unique genetic variants (Hlongwane et., 2020). However, this variation was not included when developing the PorcineSNP60 v2 genotyping BeadChip. Because of this, only 27,740 SNPs were available for analysis, although the threshold of deviation from Hardy-Weinberg Equilibrium was set to P<0.0001.
Question: This explanation does not address this concern at all.
Author Response
Reviewer 4:
Furthermore, the manuscript is challenging to understand.
Response: The manuscript was revised to improve readability. All changes are indicated in yellow.
Hence, most of the results obtained from the signatures may be estimations, not based on the actual data obtained from the pigs used in this study.
Question: Unclear. In other words, how did the authors exclude hitchhiking effects in the signatures? That is, some of the likely genes shown in the tables may include genes that are not related to population differences.
Response: To clarify the reviewer’s queries, we revised the manuscript to address the limitations of the statistical methods used. We also explained how the methods used mitigate influences such as hitchhiking effects by focusing on haplotype structures, comparisons between populations, and identifying regions that deviate from neutral expectations (Lines 121-138).
Lines 137-144: No actual number of pigs in each population is provided in the text at all. It should be shown in this manuscript.
Question: No numbers are shown in the lines. Again, please indicate the number of pigs used in each population such as LWT, SAL, ALN, ORT, MOP, etc.
Response: The revised manuscript includes a table that provides the numbers of pigs per population. Here we also highlighted the small population sizes for the Wild Boar, Bush Pig and Vietnamese Potbelly pig populations.
The small sample size in each population would not reflect all the genetic variation in the population. Hence, the three analyses, iHS and XP-EHH and HapFLK, might not work adequately.
Question: This is true only for the pigs used in the study. However, it would not be possible to detect all effects of positive selection. An explanation would need to be made for the small sample size in the Discussion.
Response: We agree with the reviewer. In the revised manuscript it is stated in the Discussion that the statistical power to detect selection signals in the Bush and Vietnamese Potbelly pigs were limited because of the small sample sizes. We clearly stated that this reduces the statistical power to detect selection signals (Lines 364-367). Also, we emphasized that further research is needed based on larger sample sizes (Lines 478-480).
Line 150: The threshold of deviation from Hardy-Weinberg Equilibrium is set to P<0.0001. This threshold is lower than thresholds for iHS and XP-EHH (both P<0.001) and HapFLK (P<0.10). Is there the possibility that SNPs at the thresholds between P<0.0001 and P<0.001, which do not deviate from Hardy-Weinberg Equilibrium, are identified as ghost SNPs in iHS, XP-EHH, and HapFLK analyses? In other words, to prevent ghost SNPs as little as possible, why is a much higher threshold (e.g., P<0.05) used for the deviation testing from Hardy-Weinberg Equilibrium?
Question: This explanation does not address this concern at all.
Response: Overall, we attempted to eliminate any spurious SNPs from our analysis, hence the strict settings to avoid such SNPs. However, we agree with the reviewer, that by employing this high level of statistical rigor, we could not exclude the possibility that some of the signatures identified represented remnants of passed genetic exchange in the admixed genomes. To clarify this issue, we therefore added a section in the Discussion (see Lines 469-477).